# Recent Advancement in Breast Cancer Research: Insights from Model Organisms—Mouse Models to Zebrafish

**DOI:** 10.3390/cancers15112961

**Published:** 2023-05-29

**Authors:** Sharad S. Singhal, Rachana Garg, Atish Mohanty, Pankaj Garg, Sravani Keerthi Ramisetty, Tamara Mirzapoiazova, Raffaella Soldi, Sunil Sharma, Prakash Kulkarni, Ravi Salgia

**Affiliations:** 1Department of Medical Oncology and Therapeutic Research, Beckman Research Institute, City of Hope Comprehensive Cancer Center and National Medical Center, Duarte, CA 91010, USA; 2Department of Surgery, Beckman Research Institute, City of Hope Comprehensive Cancer Center and National Medical Center, Duarte, CA 91010, USA; 3Department of Chemistry, GLA University, Mathura 281406, Uttar Pradesh, India; 4Translational Genomics Research Institute, Phoenix, AZ 85338, USA; 5Department of Systems Biology, Beckman Research Institute, City of Hope Comprehensive Cancer Center and National Medical Center, Duarte, CA 91010, USA

**Keywords:** mouse, zebrafish, GEM models, PDX models, CDX models, orthotopic models, drug screening

## Abstract

**Simple Summary:**

The current article provides an overview of the significance of using various mouse and zebrafish models in cancer research. In addition, this review discusses an interdisciplinary ‘Team Medicine’ approach that has aided in enhancing our understanding of carcinogenesis and establishing new therapeutic approaches. The biological course of human malignancies, preclinical research on prospective cancer treatments, and cancer prevention benefit significantly from experimental animal models. The molecular mechanisms underlying tumor growth, progression, metastasis, maintenance, and acquisition of chemo-resistance have received significant attention in the study using different types of mice cancer models. Zebrafish have been recommended as a potential model to investigate human cancer because of their suitability for in vivo imaging, fast development, chemical screening, and adaptable genetics. The zebrafish’s forward genetics and vertebrate biology make it a model system with immense potential for understanding carcinogenesis. The significance of applying various animal models in studying cancer development and progression has been well proven. The current review provides a comprehensive description of how these diverse models may be applied productively based on the scientific challenges that need to be addressed.

**Abstract:**

Animal models have been utilized for decades to investigate the causes of human diseases and provide platforms for testing novel therapies. Indeed, breakthrough advances in genetically engineered mouse (GEM) models and xenograft transplantation technologies have dramatically benefited in elucidating the mechanisms underlying the pathogenesis of multiple diseases, including cancer. The currently available GEM models have been employed to assess specific genetic changes that underlay many features of carcinogenesis, including variations in tumor cell proliferation, apoptosis, invasion, metastasis, angiogenesis, and drug resistance. In addition, mice models render it easier to locate tumor biomarkers for the recognition, prognosis, and surveillance of cancer progression and recurrence. Furthermore, the patient-derived xenograft (PDX) model, which involves the direct surgical transfer of fresh human tumor samples to immunodeficient mice, has contributed significantly to advancing the field of drug discovery and therapeutics. Here, we provide a synopsis of mouse and zebrafish models used in cancer research as well as an interdisciplinary ‘Team Medicine’ approach that has not only accelerated our understanding of varied aspects of carcinogenesis but has also been instrumental in developing novel therapeutic strategies.

## 1. Introduction

Cancer is a significant cause of mortality worldwide and is an immense financial and societal burden. The hallmarks of cancer, according to Hanahan and Weinberg, are biological characteristics that enable cancer cells to sustain proliferative signaling, evade growth suppressor mechanisms, activate invasion and metastasis, gain replicative immortality, induce angiogenesis, resist cell death, avoid immune destruction, and deregulate cellular energetics [1,2]. Although a plethora of in vitro cellular and in vivo studies involving mouse/zebrafish models have provided tremendous insight into the mechanisms that allow cancer cells to acquire these biological attributes, cancer remains a deadly disease with a poor 5 year survival rate, owing to the development of therapeutic, disease recurrence, and distant metastasis. Hence, in support of further advancing pre-clinical studies and new cancer treatments, researchers are continuously trying to refine existing or develop new models. Thus far, mouse and zebrafish models have served as critical tools in preclinical and translational research, including drug screening, assessing therapeutic efficacy, identifying biomarkers, and molecular subtyping [3,4,5]. Notably, to identify appropriate therapeutic targets, proof-of-concept experiments have been performed employing varied mouse models, including models of spontaneous and chemically induced carcinogenesis, tumor transplantation, PDX, and transgenic and knockout mice. The laboratory mouse is the most often used animal model in cancer research because of its high level of environmental adaptability, genetic variety, and physiological resemblance to humans. Numerous models have been created primarily to target significant biological processes that are responsible for cancer characteristics, such as cell proliferation, cell cycle progression, survival, apoptosis, migration, invasion, metastasis, and angiogenesis. The development of breast cancer pre-clinical models will be the main emphasis of the current review (Table 1). The most common disease diagnosed and the second largest cause of cancer-related deaths in women is breast cancer [6,7,8]. Noteworthy is the fact that one in every eight American women will eventually acquire breast cancer. The understanding of breast cancer growth and dissemination has advanced significantly, yet there are still few therapeutic options available, particularly after metastases. A wide range of breast cancer mouse models, including xenograft, genetically engineered mouse (GEM), spontaneous, carcinogen- or virus-induced models, have been created in light of the heterogeneity of breast cancer. In addition to creating genetically replica models of human breast cancer, the introduction of GEMs, such as transgenic and knockout models, has opened up new possibilities for characterizing the processes involved in cancer initiation, progression, and metastasis [6,7,8,9,10,11,12,13,14,15]. This review will provide a concise synopsis of transgenic models that have been instrumental in delineating the biochemical and molecular alterations underlying cancer, the advancement of experimental therapies, and the development of small molecule inhibitors and other approaches for the treatment and prevention of cancer (Figure 1).

## 2. Mouse Models: Genetically Engineered Mouse (GEM) Models

GEM models have significantly contributed to our understanding of molecular mechanisms underlying tumorigenesis. Additionally, they have been crucial in determining gene functions, finding fresh targets and tumor biomarkers, and evaluating fresh treatment approaches. In particular, the advancement of transgenic and gene-targeting technologies in mouse embryonic stem (ES) cells has sped up the creation of GEM models [36,37,38]. More frequently, transgenic, and gene-targeting techniques, such as knockouts and knockin models, are used to either activate oncogenes or inactivate tumor-suppressor genes in vivo to produce mice models of cancer. Gain-of-function research uses transgenic, conditional transgenic, and knockin techniques, whereas loss-of-function studies use knockout or conditional knockout alleles.

### 2.1. Transgenic Mice

The pronuclei of fertilized zygotes are microinjected with foreign DNA by researchers to create transgenic mice. The transgenic sequences are randomly inserted into various locations across the mouse genome with varying frequencies. Depending on the type of promoter or regulatory element utilized, the transgene may express itself in particular tissues at particular embryonic stages. For instance, the promoter of the mouse mammary tumor virus (MMTV) is widely known and frequently utilized to produce mouse models of breast cancer [39]. The generation of tumors in these transgenic mice demonstrates the in vivo oncogenic potential of a particular gene of interest. Mouse ES cell-based transgenesis, in which mouse ES cells are either electroporated with transgenic DNA or infected with lentiviruses harboring transgenes, is a modified version [40]. The transgenic approach offers several advantages, for example, it is a very simplified method to assess in vivo tumorigenic functions of a gene, and the generation of transgenic mice is less time-consuming compared to other gene-targeting approaches [41,42].

### 2.2. Gene-Targeting Approach

Gene targeting enables disruption or mutation of a specific endogenous locus of a gene. This strategy relies on homologous recombination in ES cells to replace chromatin with a vector that shatters the allele. It entails several stages that either result in the knockout (delete the coding sequence of a gene) or knockin (insert foreign sequences into the targeted region) of a gene [43]. In the knockout method, a reporter gene cassette (such as lacZ or GFP) or a neomycin-positive selection marker is substituted for the coding region of the target gene (exons required for the function), resulting in a null allele. Using this method, it has been demonstrated that a number of important tumor-suppressor genes, including Rb, p53, and Brca1 [44,45,46,47], are perturbed in mice. Our knowledge of the roles played by tumor suppressor genes throughout carcinogenesis and embryonic development has greatly benefited from these knockout animals. To examine Li-Fraumeni syndrome, which is brought on by p53 germ-line mutations, p53 knockout mice have been employed [45]. The lack of geographical and temporal control over the target gene, however, is a fundamental drawback of this strategy. Since oncogene or tumor-suppressor gene disruption in the mouse germ line frequently results in embryonic mortality [6,7,8,45,48,49], this is especially concerning.

### 2.3. Conditional and Inducible Systems

Tissue-specific and time-controlled somatic mutations are introduced through conditional and inducible methods. Cre-loxP is the method most often employed in mice for conditional gene expression. Site-directed DNA recombination between the two 34-base pair loxP sequences is reconciled in this method by Cre recombinase. Excision, inversion, or translocation of host DNA occurs because of recombination, depending on the relative orientation of the two loxP sites [50]. The knockin approach introduces two loxP sites with the same orientation into the specific gene locus. Cre recombinase is then created by crossing these mice with transgenic mice expressing Cre in a particular tissue or by administering a Cre-expressing lenti- or adenovirus. This results in the generation of conditional knockout or transgenic mice, which have been effective in studying the roles of otherwise embryonic lethal genes in conventional knockout mice. The Cre-loxP system can also activate an oncogene conditionally.

### 2.4. Virus-Mediated Gene Delivery

Strong genetic tools for somatic cell gene transfer in mice include DNA and RNA tumor viruses [51]. Somatic cell genetic modification caused by viruses differs from germ-line modification techniques in that the changes only affect a fraction of the cells and are not passed down unless they also affect germ cells. For instance, Cre or dominant-negative tumor-suppressor genes are often delivered to mammalian cells in vivo and in vitro using replication-deficient recombinant adenoviruses [52]. Cre-expressing adenoviruses have been utilized to create lung and ovarian cancer mouse models [50,53]. The adenovirus, however, cannot support persistent gene expression because it cannot integrate into the host genome. Retroviral and lentiviral vectors, in contrast, allow effective and reliable gene delivery to mammalian cells by integrating into the host genome. To create a mouse model of human cancer, retroviruses have been widely employed to deliver oncogenes, shRNAs, Cre, and dominant-negative tumor-suppressor genes into mice cells or tissues [54,55]. In general, GEM models have provided stable platforms for investigating the potential carcinogenic consequences of environmental variables, imaging tumors for monitoring the effectiveness of treatment over time, developing, and testing anticancer medicines, and discovering new cancer genes.

## 3. Xenograft Models

The term “xenograft” is derived from the Greek word Xenos meaning “foreign”. Xenografts are obtained from one organism and implanted into another organism (typically immunocompromised). Xenografts have been used for studying the progression of the tumor type in humans and are widely used in pre-clinical studies [56]. Xenograft models are advantageous as they are relatively inexpensive and easy to generate, and tumors appear with a relatively short latency. The classical cell-line-derived tumor xenograft model is developed by injecting cultured cancer cell lines into an immuno-deficient mouse. Nevertheless, a drawback is associated with this approach: as the cultured tumor cells are repeatedly passaged in vitro, they adapt to the external culture environment and eventually lose the parent patient’s original characteristics.

### 3.1. Subcutaneous Inoculation

The subcutaneous implantation method is used to achieve rapid tumor engraftment to perform tumor transplantation in a new animal. This procedure has helped study the process of metastasis and tumorigenesis and has benefited pharmacological studies. This method is straightforward, economical, and rapid. Briefly, cell suspension or tumor tissue is incorporated between the skin layer and the muscle. The most preferred site for inoculation is the dorsal flanks. Tumor growth, indicated by a local nodule, is typically observed within two weeks after cell transplantation. Vascular support is critical for tumor growth and influences responses to therapy [57]. In this model, therapeutic compounds present different routes of administration depending on their action pathway and effectiveness [58].

### 3.2. Patient-Derived Xenografts (PDXs)

Over the past few years, PDX models have drawn more interest in pre-clinical cancer research. PDXs have been essential in assessing the effectiveness of anticancer medications and identifying biomarkers for drug sensitivity and resistance [59,60,61,62,63]. PDXs are regenerative tumor models that may be grown in immunocompromised mice from freshly isolated human tumors without prior in vitro exposure. Regarding tumor heterogeneity, histology, gene expression, genetic alterations, and therapy responsiveness, they duplicate patients’ tumors [29,64,65] (Figure 2). As a result, they are more reliable than cancer cell lines at predicting tumor development and therapeutic response. PDX models have also been shown to preserve the hereditary stability of primary tumors at early stages in several malignancies, including B-cell lymphoma, breast cancer, non-small cell lung cancer, and neuroblastoma [63,66,67,68,69].

#### 3.2.1. Establishing PDX Models

Tumor tissue from a patient is taken and put into immunodeficient mice to create a PDX model. The first successful passage of tumor tissue is designated as F0 (or G0), and subsequent successful passages are designated as F1, F2, F3, etc. Typically, tumor tissue is collected after a biopsy or surgical removal of the tumor to create PDX models. However, it has also been claimed that tumor cells obtained from pleural effusion or ascites [70,71] are reliable for creating PDX models. When tumor fragments are used to generate the PDX model, two different pretreatment methods may be used before implantation. The first method cuts the tumor tissue into approximately 20–40 mm^3^ pieces [72], followed by subcutaneous or orthotopic implantation. In the second method, tumor tissues are processed into a single-cell suspension for subsequent implantation [73]. Each of these methods has advantages and disadvantages. Tissue fragments can retain cell–cell interactions, which mimic the tumor microenvironment; on the other hand, single-cell suspensions avoid issues with tumor heterogeneity, but have a lower implantation success rate because of chemical or physical damage during pretreatment [74,75]. For most models, fresh tumor tissue is preferred for subsequent implantation; however, frozen tumor tissue has also been reported [67]. The subcutaneous area in the flanks is the most often chosen location for implantation. This is so that the location can be rapidly assessed, tumor growth can be easily tracked, and the implantation process is cheap. Numerous studies have demonstrated that the reactions of PDX models to therapy are strikingly similar to those of patients [76,77].

#### 3.2.2. Transplantation Sites

The subcutaneous area of the mouse flank is the transplanted location that produces PDX models most frequently. The benefit of subcutaneous models is that minimal tissue injury can result from the straightforward surgery needed to produce the PDX model. As a result, the mouse may recover from surgery swiftly. The ability to immediately assess tumor development via the skin also makes it simple to confirm growth and track changes in tumor volume over time. The fact that a secondary tumor develops in a setting unrelated to the initial location or organ presents a downside in that its features diverge from those of the main tumor. Furthermore, subcutaneous models typically fail to replicate the metastatic processes [75]. To build a large PDX cohort quickly, it may be advisable to start with subcutaneous models. The orthotopic model, in contrast, makes an effort to achieve beyond the subcutaneous model’s drawbacks. By surgically implanting tumors in the same location as the parent tumor-derived organs, orthotopic models are created. The mammary glands, the tissue from which breast cancer starts in mice, are readily accessible, making it possible to transplant tissues into them without the need for extensive surgery, making breast cancer orthotopic models the most accessible [77]. Additionally, because orthotopic models are implanted in the same organs as original tumors, they maintain the microenvironment features of those primary malignancies and, therefore, are better suited for investigations on metastasis [75]. Although, to properly implant tumor tissue requires expert surgical procedures. Additionally, there is a constraint that demands employing imaging technologies like computed tomography or ultrasound to monitor tumor growth because it is typically impossible to detect tumor growth just by palpating the mouse from the outside. The fact that PDX models preserve the original tumor architecture, including cellular and histopathological structures, is a significant benefit. Overall, the pathohistological, genetic, and therapeutic susceptibility of tumors in PDX models is comparable to those of the parent tumors. In fact, clinical information from the patients from whose PDX models were produced corresponds with the PDX models’ susceptibility to anticancer medications. Additionally, cytogenetic examination of tumor cells from PDX models has shown that the genomic and gene expression patterns between the models and the original patient tumors have been significantly preserved [63,78]. To study several aspects of tumor biology, including cancer development, mortality, evolution, and metastasis, PDX models can be used. A growing body of research indicates that PDX models are quite good at forecasting the effectiveness of both traditional and cutting-edge anticancer therapies. They are starting to be recognized as useful examples of translational research that can help with precision medicine [79]. Intact female NSG mice are utilized without estrogen supplementation for PDX formation and maintenance since endogenous estrogen levels are adequate. The formula 4/3 × r1 × r2 (r1r2) [0.125], where r1 is the smaller radius, is used to compute tumor volumes and to evaluate tumor development. Usually, mice are asphyxiated with CO_2_ after the tumor reaches a size of around 1000 mm^3^. Liquid nitrogen is used to quickly freeze the tumors after removal. In order to conduct immunohistochemical analyses on tumors, they are kept at −80 °C until biochemical tests.

#### 3.2.3. Mouse Strains Used for Developing PDX Models

The success rate of developing PDX models is associated with the mouse strain in which tumor tissue is implanted. The most commonly used strains for developing PDX models include nude, NOD-SCID, and NOD/SCID/interleukin-2 receptor gamma chain null (NSG) mice [80]. A nude mouse lacks a thymus, which results in a significantly lower quantity of T lymphocytes. They do, however, possess functioning B and NK cells, which makes it difficult for original human tumor tissue to proliferate. The success rate of PDX models created in nude mice is hence unacceptable. However, due to the lack of functioning T or B cells, NOD-SCID and NSG mice suffer a more severe immunodeficiency than nude mice. In contrast to NOD-SCID mice, NSG animals have a complete deficiency in NK cells [29,81]. Therefore, the best mouse strain for creating PDX models has been determined to be NSG mice (Table 1).

#### 3.2.4. PDX Model for Breast Cancer

Breast PDX models are created from human tumor samples an hour after they are obtained. The 1% penicillin/streptomycin supplemented DMSO/high glucose medium is used to collect tumor tissue that has been frozen. Sterile conditions are used for every process. Using a scalpel or razor, tumors are cut into pieces of 2–3 mm, which are then placed in the Matrigel (Corning, Corning, MA, USA) medium [1:1]. After depilating the mouse surgery site with hair removal lotion, Techni-Care is applied to disinfect it (Care-Tech Laboratories, Inc., St. Louis, MO, USA). Tumors were surgically placed into the inguinal fourth mammary fat pad of female NSG mice aged 6 to 8 weeks [79]. Comprehensive genomic investigations have shown that PDX models sustain the same overall global gene expression and activity as the parent tumors, and PDX models appear to preserve the genetic properties of their original tumors [82,83]. In order to bridge the gap between laboratory discoveries and clinical translation, these mice models are, therefore, regarded as effective for pre-clinical research of focused therapy methods and molecular analyses.

#### 3.2.5. Humanized PDX Models in Cancer Immunotherapy

Immunotherapy is potentially an effective modality for treating cancer because the immune system plays a crucial role in both promoting and preventing of tumor growth [84,85]. However, to avert the rejection of transplanted human tumor tissue or cells, PDX models are generated in immuno-compromised mice. However, these models cannot be used to investigate the immune system’s interactions with the tumor and related milieu, including immune cell invasion. To circumvent this, an array of humanized mouse strains, including genetically manipulated and immunologically humanized mice, have been developed [30,86,87]. Human immune cells were grafted onto mice to create immunologically humanized mice, and the murine gene was substituted with a human transgenic counterpart to create genetically modified humanized mice [85]. Thus, in light of this, immunologically humanized mice can be employed for the construction of PDX models.

## 4. Orthotopic Models

In order to evaluate the effectiveness of innovative anticancer therapy methods, especially for metastasis, orthotopic xenograft models are frequently utilized. These models are free from the various restrictions of subcutaneous xenograft models.

### 4.1. Use of Low-Molecular-Weight Heparin to Decrease Mortality in Mice after Intracardiac Injection of Cancer Cells

Research using mouse models that mimic the metastasis of human tumors to bone is critical for developing anticancer therapeutics. Breast, prostate, lung, kidney, and thyroid cancer patients are more prone to bone metastases [88,89,90,91]. However, in mouse models of naturally occurring breast and prostate cancers, artificially implanted animal tumor models (such as syngeneic and xenograft tumors), and models involving chemical or transgenic production of mammary and prostate cancers, bone metastasis is uncommon.

To overcome this, intracardiac injection of human tumor cells into anesthetized nude mice is used to establish a bone and brain metastasis model. However, intracardiac injection of some human tumor cell lines causes acute, detrimental neurologic effects and high mortality. Numerous investigations also revealed that low-molecular-weight heparin (LMWH; enoxaparin) pretreatment limits the hypercoagulable condition that can be brought on by an intracardiac injection of tumor cells and prevents platelet consumption and thromboembolic development. In addition, intravenous enoxaparin injection before intracardiac injection with breast carcinoma lines, such as MDA-MB231Br-Luc, dramatically decreased mouse mortality while still permitting the development of brain metastases [9]. Therefore, using LMWH in mice is expected to prevent morbidity and mortality associated with intracardiac injection of human tumor cell lines.

Furthermore, thromboembolism can occur in intracardiac tumor-challenged mice and LMWH can block thromboembolism. Therefore, the mortality reduction by pretreatment with LMWH increases the types of cells that can be studied using the intracardiac injection metastasis model and decreases the number of mice required for these studies. For example, when mice are given 10 mg/kg enoxaparin intravenously before injection of 0.1 × 10^6^ MDA-MB231Br-Luc cells, they can typically expect 95% survival. Enoxaparin injection does not produce clinically observable adverse effects, and almost all mice survive until the study’s end. Additionally, pretreatment with enoxaparin has no discernible effect on the tumor burden since both pretreated and untreated animals exhibit identical amounts of bioluminescence. As a result, 10 mg/kg enoxaparin given intravenously 10 min before an intracardiac tumor cell challenge can reduce mortality linked to MDA-MB231Br-Luc cells without significantly changing tumor burden. These findings show that intracardiac injection of MDA-MB231Br-Luc cells causes a hypercoagulable condition that leads to platelet consumption, widespread pulmonary thromboembolic development, and mortality. Enoxaparin pretreatment stops the onset of this hypercoagulable condition.

### 4.2. Intracardiac Injections

Prior to the intracardiac injections, mice are intraperitoneally given 90 mg/kg of ketamine and 10 mg/kg of xylazine to make them unconscious. Using a 0.5 mL tuberculin syringe and 27-G needle, luciferase-tagged MDA-MB231Br-Luc breast cancer cells in 100 µL serum-free medium are injected into the left ventricle of dorsal-reclining mice. Mice are observed using bioluminescence imaging five minutes after intraperitoneal injection of 150 mg/kg d-Luciferin. d-Luciferin interacts with the luciferase-tagged tumor cells and correct intracardiac administration results in the dispersion of bioluminescence throughout the animal. Any mice displaying neurologic clinical indications should be put to death, and animals should be watched until they recover from anesthesia.

### 4.3. Imaging

The bioluminescence of human tumor cell lines expressing firefly luciferase and green fluorescent protein is detected using in vivo BLI to track the development and extent of metastasis. d-luciferin [150 mg/kg i.p.] is injected into mice, and they are then given 3% isoflurane in oxygen to make them unconscious. Dorsal and ventral photos are taken of mice when they are under the effects of isoflurane anesthesia 10 min after luciferin injection. After the first week, until the study’s conclusion, images are gathered twice weekly. Images are gated for areas of interest, such as the entire body, both hind limbs, and the area around the head, and are then examined using software called Living Image (Xenogen, Hopkinton, MA, USA). If the luminous signal rises over a modest minimum threshold that has been deliberately set, the area is regarded as tumor positive. The mean number of photons per second is determined each time in order to quantify in vivo bioluminescence.

## 5. Developing Inventive Therapeutic Strategies and Imaging Techniques Using Transgenic Animals

The use of human tumor xenografts, animal allografts, or tumor tissue xenografts grown in immunocompromised mice has been made for the in vivo study of possible anticancer medicines. The milieu around the injected tumor cells in the majority of xenograft techniques does not accurately reflect the microenvironment that normally surrounds a growing tumor. GEM models may offer a supplementary system that overcomes some of the limitations of xenograft models in drug development. Transgenic mice have been used to assess the chemoprevention, therapy, and metastasis inhibition effects of several medicines. These animals have been crucial in evaluating the effectiveness and mechanism of action of chemotherapy drugs that are thought to either directly or indirectly target the oncogene. For instance, cyclooxygenase 2 (COX-2) is overexpressed in mammary tumors from MMTV-neu mice and HER-2/neu positive human breast malignancies [92].

### Imaging Using Transgenic Animals

In order to identify tumors and metastases, as well as to non-invasively monitor the success of treatment interventions in cancer patients and the efficacy of new medications in pre-clinical animal models, a number of imaging methods have been developed. In both pre-clinical and clinical applications, anatomical and physiological information on tumors can be obtained using computed tomography, magnetic resonance imaging (MRI), and ultrasound (US). In addition, the use of positron emission tomography and single-photon emission computed tomography can provide imaging information at the molecular level. Other imaging modalities include fluorescence reflectance imaging, fluorescence-mediated tomography, bioluminescence imaging [BLI], laser-scanning confocal microscopy, and multiphoton microscopy. Details on these imaging modalities have been reviewed in Condeelis and Weissleder [93]. For example, the progression of mammary tumors in MMTV-*Neu* (activated *neu*) transgenic mice was followed using MRI and ultrasonography [94]. US imaging helped obtain basic information such as the size of developing tumors and identifying necrotic areas. On the other hand, advanced analysis of morphological aspects was possible with MRI, providing high-resolution images that could be used for differentiating details of necrotic areas such as coagulation, liquefaction, biphasic splitting of cysts, and fibrotic and lipidic infiltration [94].

## 6. Zebrafish Models: Zebrafish Xenograft Models

Extensive testing in animal models is necessary for the discovery and development of novel anticancer medications in order to determine the safety and efficacy of therapeutic candidates [95]. As mentioned above, tumor xenografts, or the transplanting of human tumor tissue into mice, are frequently employed to monitor cancer development and evaluate the anticancer effectiveness of prospective medications. However, because of the lengthy timeframes required for tumor formation in mice models, it is not feasible to evaluate a large number of medication candidates quickly. Furthermore, prolonged growth times increase the likelihood that tumors will gain genetic and epigenetic changes. In contrast, tumor xenograft studies in zebrafish provide an efficient platform for rapid testing of the safety and efficacy of anticancer agents in less than two weeks (Figure 3). The unique features of zebrafish that enable patient-specific chemosensitivity analyses include speed (5–7 days) and small patient tissue requirements (100–200 cells per animal). In addition, imaging of the small, transparent fry is unparalleled among vertebrate organisms. Thus, the zebrafish xenograft assay is ideal for evaluating new agents, including in the context of personalized medicine [96]. Rapid, trustworthy, and pertinent biological models are necessary to screen and validate drug candidates for both effectiveness and safety while creating novel cancer treatments. Zebrafish have become a top model organism for these objectives in recent years. To assess prospective therapeutic options, human cancer cells can be engrafted into larval or immunocompromised adult fish. Given that zebrafish and humans share 80% of disease-related orthologous genes, they offer a high-throughput, low-cost alternative to mouse xenografts that is also relevant to human biology [4,5]. The background information about zebrafish xenograft models’ procedures and applications in cancer research is provided here [97].

### 6.1. Zebrafish as a Model for Studying Human Cancer

Finding novel cancer treatments requires both in vivo testing in animal models and in vitro testing and validation in human cell models. Several potential treatments may be promptly tested against different cancer types using cell models. Traditionally, GEM models or xenografts of human cancer cells using immunocompromised mice have served as the gold standard for evaluating the safety and efficacy of anticancer treatments in vivo (e.g., NOD-SCID). However, mouse-based studies are time-consuming and frequently inappropriate for high-throughput screening and toxicity evaluation of anticancer medications. Zebrafish have proven themselves to be a reliable tool for toxicity testing and drug discovery [98,99]. Zebrafish constitute an appealing model for the development of cancer drugs due to the variety of traits they exhibit. They are far more fertile than mice. Successful mating can result in hundreds of fertilized eggs, which hatch at around 72 h after fertilization and quickly grow from embryos into larval fish [100]. At 90 days after fertilization, zebrafish are ready to reproduce. Zebrafish are not mammals, yet their vertebrate architecture is similar to that of humans, and their genome has orthologs for 70% of human proteins [101]. The creation of transgenic zebrafish models with altered gene expression is quite simple with current technology. Numerous malignancies have been studied using transgenic zebrafish [102,103,104] and transgenic models with fluorescent proteins expressed only on vascular endothelial cells have led to new understandings of neo-angiogenesis and tumor-induced vascularization [104].

### 6.2. Embryo-Larval Zebrafish Xenografts

Zebrafish are a viable model for the transplanting of human cancer cells [105,106,107]. Human cells that have been transplanted may move, live, and interact with their new environment. The study of multiple human cancer lines, including those from melanoma, breast, and leukemia, among many other cancer types, has been made possible through zebrafish xenograft research [108,109,110]. Zebrafish have innate immune cells, but they do not have an adaptive immune system until 30 days after conception, which makes embryo-larva zebrafish xenograft models advantageous [111]. This desirable property disallows the use of immunosuppressive medicines or immunocompromised variations in xenograft procedures and enables the xeno-transplantation of human cancer cells without immune rejection. Zebrafish may also be housed in groups in petri dishes or individually on 96-well plates due to their small size, which makes handling and upkeep simple. In addition, zebrafish can live in temperatures between 32 and 36 °C, which is closer to the conditions used in human cell culture [112,113,114]. Zebrafish prefer an ambient temperature of 28 °C. Furthermore, zebrafish xenografts may be established with only a few hundred cells, compared to thousands for mouse xenografts. This is crucial when cancer cell populations are limited, as they are when cancer cells are isolated from primary patient tissue samples. Additionally, zebrafish embryos have a full complement of orthotopic organs and tissues, including the brain, heart, and liver, as well as a working circulatory system, by about two days after fertilization [100,115].

Zebrafish are more accessible for in vivo testing than mammalian models because they require less upkeep and care. As zebrafish larvae and embryos are transparent, imaging tools often used by researchers, such as common epifluorescence and confocal microscopes, can be used to non-invasively examine cancer growth and possible medication effects within the host. The capture of fine pictures of fluorescent cancer cells was made possible by the translucent zebrafish tissue’s exceptional optical penetration. The excitation/emission spectra of fluorescent probes also pass-through tissue with low light scattering because zebrafish tissue is micron thick, especially at longer wavelengths. It has been successfully used to visualize not only individual cells in zebrafish but also subcellular structures like centrosomes, endosomes, mitochondria, microtubules, etc., using fluorescent proteins like mCherry and green fluorescent protein (GFP) and fluorescent dyes like CM-DiI and 5-chloromethylfluorescein diacetate (CMFDA) [116,117,118,119]. Numerous human cancer cell lines may survive and multiply in the zebrafish yolk, including those for neuroblastoma, melanoma, leukemia, breast, prostate, and ovarian cancers [107,108,109,110,111,120,121,122,123,124,125,126]. The most common objectives are growth, survival, invasion, and metastasis. When cancer cells are implanted into the yolk, several of their properties may be evaluated. Rapidly migrating aggressive cancer cells can leave the yolk sac and spread throughout the organism via circulating in the circulation [127]. Non-metastasizing cell lines did not spread to zebrafish; however, breast, prostate, colon, and pancreatic cancer cell lines did [128]. Additionally, studies using zebrafish xenografts have confirmed the findings of those carried out using mouse xenografts.

The formation of tumor blood vessels is crucial. Tumor cell and blood vascular interactions have been studied using zebrafish. Angiogenetic substances are released by tumor cells to promote the development of blood vessels [129]. Zebrafish that have vascular endothelial cell-specific GFP expression have been created [104,108,130,131] in order to evaluate tumor-induced angiogenesis. The fli1a: EGFP [132] and kdrl:EGFP [133] lines are two of the most widely used GFP-expressing zebrafish lines utilized for researching developmental and tumor-induced angiogenesis. Zebrafish blood arteries may be observed expanding toward transplanted human tumor cell masses within 24 h of cell implantation, providing a measure of angiogenesis [108,134,135]. Zebrafish blood arteries provide the required architecture to evaluate tumor cells’ metastatic traits, such as extravasation and intravasation. Cancer cells must intravasate into the circulation and extravasate out of it in order to spread to different organs and tissues during metastasis [136]. When implanted into the circulation of zebrafish, tumor cells have the ability to extravasate by adhering to the endothelium and leaving the capillaries [137]. As tumor cells injected into the yolk can enter the circulation, go to the tailfin, and create micro-metastases, intravasation has also been observed [128]. Important discoveries regarding the mechanics of sprouting angiogenesis [138,139,140,141], vessel guidance [135,136,137,138,139,140,141,142], and vascular endothelial growth factor signaling [143,144], among other angiogenic processes [145], have been made thanks to the use of zebrafish. Time-lapse imaging can reveal information on the activities of cancer cells and suggest pathways for invasion, proliferation, and metastasis. Overall, zebrafish xenografts can perceive and picture a tumor-like environment in vivo.

### 6.3. Zebrafish Xenografts for Cancer Drug Screening

The embryogenesis of humans and zebrafish is comparable, with gene expression across phyla being preserved and many homologous anatomical and physiological structures and functions [99,146]. Additionally, the small size of the embryo-larva zebrafish makes it easier to test for cancer-related drugs since one or more zebrafishes may be kept in 96-well plates at this stage. The 96-well format facilitates quick and simple treatment using various quantities of various small compounds [147,148]. Treatments are administered directly to the fish medium, allowing for a fast assessment of systemic toxicity. Reduced survival or phenotypic abnormalities are two toxic outcomes that are simple to spot, and other developmental and behavioral endpoints may be observed without causing harm [147]. Additionally, each embryo-larva only needs a few microliters of the medium, and fish may be easily photographed with a wide-field objective, allowing for quick evaluation of phenotypic changes. Data on big xenografts may be acquired fast using high-content microscopes [149,150]. When employing zebrafish xenografts, a variety of techniques may be employed to evaluate changes in cancer growth. Transplanting fluorescent cancer cells into zebrafish embryos and letting the fish grow and mature over a few days is the first step in each procedure. Then, to compare treatment groups, fluorescent cells are fixed and counted using a hemocytometer or flow cytometry [108,123]. This method of removing and counting cells from dead zebrafish allows for a more precise assessment of the cancer cells’ rate of growth. Additionally, both living and dead cells as well as the amounts of protein expression can be counted utilizing various staining techniques employing fluorescent antibodies. This is significant because, due to the nature of the microinjections, each zebrafish xenograft may include a variable number of transplanted cells. Due to the tiny cell population that is engrafted during zebrafish xenograft transplantations, even modest changes in the number of transplantations can result in large percentage disparities across xenografts.

### 6.4. Promising Outcomes from Zebrafish Models

PDX models offer a more realistic method of cancer research, as was previously described. Zebrafish can act as hosts for tumor cells and tissues taken from patients, much like mice models can. In fact, PDX models that use zebrafish as “avatars” to quickly assess possible treatments are becoming more and more attractive tools [34,151]. It has been demonstrated that the behavior of the cancer tissues employed in zebrafish PDX models is comparable to that of human tumors. In zebrafish xenografts, for example, transplanted bone metastatic tumor cells from a woman with breast cancer moved to caudal hematopoietic organs that are similar to human bone marrow [152]. The successful implantation of pancreatic tumor tissue into the yolk of an embryonic zebrafish served as proof that full patient tissue may also be transplanted. Similar to the patient, this transplanted tissue exhibited metastatic tendencies [110]. Biopsy samples may be handled and transplanted into zebrafish in the same way in order to decide on the best pharmacological treatment for a certain patient. Fewer cancer cells are required to successfully develop xenografts in zebrafish embryo larva compared to standard mouse models, allowing for the production of many more xenografts from a single patient’s tumor. Therefore, zebrafish PDX models have the ability to offer healthcare professionals unique trial data on medications to enable the selection of a successful individualized treatment plan. Patients with aggressive tumors require prompt access to suitable, effective therapy; therefore, the capacity to swiftly provide this tailored data using zebrafish can dramatically enhance care and lengthen patient survival. In conclusion, Zebrafish xenograft models provide a strong platform for observing cancer cell growth and vetting potential cancer therapy choices. Zebrafish provide a vertebrate architecture with a supportive in vivo environment that includes key components, such as extracellular matrix and flowing blood arteries, required for the formation of human tumors (Figure 3). Zebrafish are reasonably priced in vivo drug testing models. They provide us with the tools to evaluate the potency and toxicity of possible medications. Zebrafish can also provide high-throughput testing, which is not possible with mouse models. Zebrafish will play a key role in the development of new cancer drugs as technology and methodology develop, acting as a link between in vitro and in vivo investigations.

## 7. CRISPR/Cas9 Platform: An Advanced Genome Editing Technique

A powerful genome editing technology called the CRISPR/Cas9 system is extensively employed in biomedical research. However, a number of issues, including off-target effects and a lack of simple solutions for multiplex targeting, continue to restrict its uses. By permitting pulse exposure of the genome to the Cas9/sgRNA complex, the development of the inducible CRISPR/Cas9 system significantly lowered off-target effects. Furthermore, by co-expressing a number of single guide RNAs (sgRNAs) with various direct sequences, the CRISPR/Cas9 system may be programmed to silence several genes at once. The development of the CRISPR/Cas9 system during the past few years has transformed functional genetics research and genome editing methods [153]. A 20-nucleotide direct sequence and an RNA scaffold called an RNA chimera sgRNA can be used to instruct the Cas9 endonuclease to create double-strand breaks in a particular DNA region. Non-homologous end-joining is a mechanism used by mammalian cells to repair double-strand breaks that frequently result in indels, or tiny insertions or deletions. Thus, by altering the reading frame or splicing locations, the CRISPR/Cas9 system offers a straightforward method to prevent the production of certain proteins. One of the most prevalent in vivo models for therapeutic target validation and pre-clinical drug testing in translational cancer research is human cancer cells produced as subcutaneous xenografts in immunodeficient mice. These cells are frequently designed for induced expression or repression of a certain gene of interest to enable a more accurate evaluation of its function(s) [153,154,155]. For instance, inducible gene expression/suppression based on the use of doxycycline (Dox) regulated tetracycline (Tet) systems offer a potent and often employed technique for functional research on the effects of gene overexpression/downregulation [83]. Furthermore, by successfully using CRISPR-Cas to target cancer driver mutations in vivo, oncogenic driver regulatory and functional pathways may be studied.

## 8. Perineural Invasion (PNI) in Breast Cancer

Perineural invasion (PNI) in breast cancer refers to the infiltration or spread of cancer cells encompassing nerves in the breast tissue. Breast cancer cells can invade the perineural region, where they can then spread through the nerve fibers and beyond the main tumor site [156]. Perineural invasion is considered a poor prognostic factor and is usually linked with aggressive forms of breast cancer, such as larger tumor size, higher grade tumors, increased risk of recurrence and distant metastasis [157]. Treating and managing breast cancer with perineural invasion is multidisciplinary, including surgery, radiation therapy, chemotherapy, hormone therapy, or targeted therapy [158,159].

The mechanisms and factors contributing to perineural invasion in breast cancer are not fully understood. Some theories rely on the fact that growth factors are released by cancer cells that attract and stimulate nerve invasion. The neurotrophin nerve growth factor (NGF), released by the cancer cells has been demonstrated to be a potential driver of tumor neurogenesis [160]. In another breast cancer study, the NGF was described as a mitogen and stimulated breast cancer cell survival via NGF receptor P140^trKA^ and P75^NTR^ [161]. Tan et al. conducted a comparative computational study to analyze the damage induced by perineural invasion for TNBC and non-TNBC, and they reported that TNBC has substantially more up-regulation of neural genes than non TNBC [162].

### 8.1. Role of Animal Models in Studying the Perineural Invasion

In vivo animal models are important in studying the perineural invasion of breast cancer, as they provide a controlled and reproducible environment to investigate the mechanisms, progression, and treatment of this occurrence [163]. Some commonly used animal models include:

Xenograft models: These models involve the transplantation of human breast cancer cells or tissue into immunodeficient mice. Human breast cancer cells can be injected into the mammary fat pad or near peripheral nerves to simulate perineural invasion. This allows researchers to monitor tumor growth, invasion of nerves, and evaluate therapeutic interventions.

Orthotopic models: In orthotopic models, human breast cancer cells are injected directly into the mammary gland of immunocompetent mice. This closely mimics the microenvironment of breast cancer and allows for the study of perineural invasion within the context of an intact immune system.

Transgenic models: Genetically engineered mouse models (GEMMs) can be developed to express specific genes associated with breast cancer development, progression, and perineural invasion. By manipulating the expression of these genes, researchers can investigate the molecular mechanisms underlying perineural invasion.

Nerve-targeted models: In these models, nerves are surgically manipulated to induce perineural invasion by injecting breast cancer cells into or near the nerves. For example, the sciatic nerve or other peripheral nerves can be exposed and injected with cancer cells, allowing researchers to study the interactions between cancer cells and nerves.

Imaging models: Advanced imaging techniques such as magnetic resonance imaging (MRI) and positron emission tomography (PET) can be utilized to visualize and track perineural invasion in animal models. These techniques enable non-invasive monitoring of tumor growth, nerve infiltration, and response to treatment over time. These animal models provide valuable insights into the complex processes involved in the perineural invasion of breast cancer. They allow researchers to investigate the underlying mechanisms, test potential therapeutic approaches, and evaluate the efficacy of anticancer drugs targeting perineural invasion. It is important to note that while animal models provide valuable preclinical data, findings must be further validated in human studies to ensure their clinical relevance.

### 8.2. Benefits of In Vivo Models for Studying Perineural Invasion

Recapitulation of complex tumor microenvironment: In vivo models provide a more precise representation of the tumor environment than in vitro cell culture methods. They allow for the investigation of the interactions between breast cancer cells, nerves, immune cells, and other components of the tumor microenvironment, which contributes to the comprehension of the complex processes included in the perineural invasion.The use of animal models facilitates real-time monitoring of tumor development, progression, and invasion. Researchers can monitor the spread of cancer cells along nerves, study the temporal and spatial dynamics of perineural invasion, and investigate the mechanisms behind this process.Evaluation of therapeutic interventions: Prospective treatment strategies that target perineural invasion can be evaluated using in vivo models. Researchers can assess the efficacy of anticancer drugs, radiation therapy, or surgical procedures in limiting nerve infiltration and preventing the spread of cancer cells to surrounding tissues.Animal models can be used to study host responses and systemic effects. These host responses and side effects are connected to perineural invasion. These models provide insight into how immune response, inflammation, neuroplasticity, and modifications in neural signaling are impacted by the presence of cancer cells invading nerves.

### 8.3. Limitations of In Vivo Models for Studying Perineural Invasion

Species differences: Animal models may not fully reflect human disease because of the innate biological differences between species. Due to the variations in human breast cancer cells’ reactions and interactions with the environment and nervous system, findings from animal models are frequently not directly transferable to clinical settings.Logistical and ethical constraints: The use of animal models raises ethical questions and caring for them and employing them in research can be expensive. Additionally, there may be a time limit on the amount of time required for the formation and study of tumors in animal models, which could limit the number of repeat trials and their efficacy.The intricacy of the human tumor microenvironment is frequently simplified by in vivo models, despite its benefits. They could limit the application of findings in the clinic since they might not completely duplicate the variety of cell types, stromal components, and molecular interactions seen in human breast tumors.Difficulty in analyzing distant metastasis: In vivo models are typically more helpful in analyzing local invasion and the early stages of perineural invasion. Investigating distant metastasis in animal models, which occurs in advanced stages of breast cancer, can be challenging because of variances in organ microenvironments and species-specific traits.

## 9. Applications of Mouse Models of Cancer

To better understand tumor biology, mouse models are very helpful, and they have shed light on the following issues:(a)What genetic changes are the initial ones that lead to the development of cancer?(b)How do cancer genes collaborate at various phases of tumor development?(c)Which cell is the source of different tumor types?(d)Why do people with the same form of cancer have varying susceptibilities?(e)What genetic modifications are there?(f)How do tumor cells multiply and spread?(g)Can the environment contribute to cancer?(h)What interactions occur between nearby healthy stromal cells and tumor cells?(i)What are the underlying chemoresistance mechanisms?(j)What processes underlie the dormancy and recurrence of tumors?(k)Which therapy approaches are effective against particular cancer types?(l)How may cancers be found in their earliest phases of growth?

The applications for which mouse models are utilized to provide answers are highlighted below.

### 9.1. Defining the Roles of Environmental Factors in Tumor Development

Environmental factors such as hormones, nutrition, UV emissions, and toxins have causative linkages to particular human malignancies. However, it is not known if these variables may directly begin or encourage tumor growth. Testing the involvement of environmental elements in tumor formation is directed by a critical use of mouse models, which is impractical or unethical to conduct on people. Tamoxifen, for instance, inhibits the development of mammary tumors in animals following oophorectomy and estrogen withdrawal, which facilitated its approval for the treatment of human breast cancer [164]. C3H inbred strains that spontaneously generate hepatomas have shown the benefit of caloric restriction in lowering the occurrence of tumors [165]. Studies on mice also showed the causal effects of UV and sunburn in melanoma [166,167]. Through the use of mice models, several substances have been linked to the development and spread of tumors.

### 9.2. In Vivo Imaging

In vivo imaging methods have increased the usefulness of mice models for tumor biology and pre-clinical research. Noninvasive tumor imaging enables the sequential measurement of a number of variables, including the results of potential treatment medications. For imaging mouse tumors, several techniques have been developed, including ultrasound, computed tomography, magnetic resonance, imaging intravital microscopy, microcomputed tomography, single-photon emission, micro-positron emission tomography, bioluminescence imaging [BLI], and whole-body fluorescence imaging [168]. The creation of several “biosensor” reporter mice has been made possible by transgenic technologies in addition to the advancement of imaging modalities. Since these mice may be utilized for positron emission tomography imaging, BLI, and fluorescence imaging, respectively, reporter mice expressing luciferase, fluorescent protein, or HSV-TK, separately or in grouping, have been most often created. Gene-targeting or transgenic methods can be used to insert luciferase or fluorescent proteins into the mouse genome after fusing them with tissue-specific promoters, transcription factors, or response regions. These reporter mice can be used as biosensors to identify the expression of oncogenes or tumor suppressor genes and to observe certain in vivo tumorigenic processes. A combination approach utilizing reporter mice and cancer mouse models can be used to monitor spatiotemporal tumor growth. The use of reporter mice, cancer mouse models, and imaging technology is projected to considerably aid future testing of possible anticancer drugs [169,170].

## 10. Promising Outcomes from Mouse Models

Animal experimental models are crucial tools for studying the biological progression of human malignancies, performing pre-clinical research on potential cancer treatments, and preventing cancer. To simulate different malignancies in patients, subcutaneous or orthotopic cell-derived tumor xenograft models (CDX models) have been developed over time. However, CDX models have two important drawbacks. One model poorly stimulates the vascular, lymphatic, and immunological microenvironments found in renal malignancies. The other model loses genetic heterogeneity compared with the corresponding primary tumor. To overcome these limitations over the past several years, PDX models have emerged as promising translational research tools. These models can retain the genetic and histological stability of their originating tumor at limited passages and thus shed light on the following areas important for precision cancer medicine [171]:Cancer genes frequently contribute significantly to healthy physiology and development. In mice, severe tissue abnormalities or embryonic mortality frequently arise from the loss or activation of tumor-suppressor genes and oncogenes [45,46,47,48]. The expression of crucial developmental pathways including the Wnt/-catenin and sonic hedgehog signaling pathways is also aberrant in many malignancies [172,173].Understanding the abnormal growth of cancer involves familiarity with how tissue normally develops.Oncogenes or tumor-suppressor genes that function in the same or a similar pathway are typically inactivated to reduce embryonic mortality in tumor-suppressor gene knockout animals or oncogene knockout mice. This can be demonstrated, for instance, in the capability of BRCA1 mutant mice to counteract the embryonic mortality brought on by the loss of p53 activity [174,175]. It is possible to explain why patients with BRCA1-associated malignancies have a significantly higher prevalence of p53 mutations than patients with spontaneous variants of the same tumors by considering this rescue, which suggests that a cell must lose the ability to function p53 to tolerate the loss of Brca1 function [176].Particular oncogenes and oncogenic signaling pathways are selectively activated in a subset of malignancies [177]. According to mice models, certain malignancies must be maintained by the persistent expression of specific oncogenes such as H-ras, K-ras, and cmyc [178,179,180].Oncogenes and tumor-suppressor genes both have context-dependent activities. Several oncogenes have multiple functions that, depending on the genetic background, can either induce or repress carcinogenesis [181,182].For survival and proliferation, tumor cells stimulate angiogenic, hypoxic, and metabolic pathways [183,184,185]. Additionally, one of the ways that tumor cells develop chemoresistance is by activating alternate survival pathways to make up for the damaged route (for example, after receiving VEGF2-targeted treatment, tumor cells stimulate VEGF-independent angiogenesis driven by FGF) [186].The most efficient medication can be suggested prior to patient therapy by evaluating a number of chemotherapeutic medicines in PDX models. Thus, PDX models offer a powerful substitute for a number of steps of the drug development process in precision oncology, including drug efficacy testing, drug resistance research, biomarker identification, and co-clinical trials.

## 11. Cancer Systems Biology

From the foregoing, it is obvious that a comprehensive, systems-level, perspective employing animal models can provide a deeper understanding of the disease pathology as well as new therapeutic strategies. Therefore, a concerted effort of a team of scientists from different disciplines including physics, mathematics, cancer biology, ecology and evolution, and clinical oncologists working together—the ‘Team Medicine’ approach—can leverage the advantages offered by these model organisms. For example, this approach can help develop better imaging techniques in live animals, identify new targets in zebrafish in a high throughput manner, and/or biomarkers employing spatial transcriptomics and machine learning algorithms, and novel treatment strategies such as ‘intermittent’ therapy in mice xenografts aided by mathematical models based on evolutionary game theory [187].

## 12. Potential Impact

For a long time, mouse models have substantially aided our comprehension of the biology and molecular mechanisms behind the spread of cancer. It has been possible to define the functions of several transgenes in the development, progression, and metastasis of cancer thanks to the capacity to deliberately overexpress/silence a single transgene. It is also possible to assess how the transgene(s) are expressed in relation to hormones and changes in genomic instability. Understanding the processes behind chemotherapeutic resistance, tumor dormancy, and tumor recurrence has been made easier thanks to transgenic mice. The effectiveness of novel treatment therapies can also be tested in transgenic animals. Finally, the development of innovative imaging modalities that can assist in early cancer diagnosis through preliminary testing in transgenic animals would be very beneficial in detecting early-stage malignancies in people, making therapeutic intervention feasible before metastatic spread. Future and present mice models will aid in creating better ways to spot cancer early on, treat it, and perhaps even stop it from spreading to other parts of the body.

## 13. Limitations and Challenges

The variations between species provide the biggest obstacle to employing mouse models of human malignancies and bringing the findings of these investigations to the clinic [188]. The size, lifespan, organ shape, and physiology of mice are different from those of humans. Telomerase activity, which is primarily inactive in mature human cells, is one significant distinction between humans and mice. The majority of mouse cells have active telomerase, which makes them more likely to immortalize and undergo transformation than human cells. Therefore, compared to human tumors, mouse tumors require fewer genetic mutations to undergo malignant transformation. Additionally, in mice, telomerase activity precludes the modeling of genomic instability in malignancies from humans. Telomerase inactivation may be required in animal models to precisely mirror human malignancies. For instance, simultaneous Terc and p53 deletion in mice leads to chromosomally uneven tumors that are more like human malignancies [189]. The histology and spectra of mouse tumors differ from those of humans because of these species-specific characteristics. For instance, p53 null mice preferentially develop sarcomas [45,48,49,190], but Li-Fraumeni syndrome people primarily develop carcinomas [191]. Further evidence that the processes generating metastasis may vary between species comes from the fact that mouse models of cancer tend to develop very few metastases or have metastases with differing tissue specificity from human tumors. Finally, a distinct pharmacological response in mice models may be caused by variations in metabolic rate and routes (for example, the cytochrome P450 pathway for drug metabolism) [192]. When mouse models are utilized for drug research and pre-clinical studies, this is especially concerning [193,194,195,196].

Since mice and humans differ in how drugs are metabolized and how well they bind to target proteins, as was previously noted, it may be difficult to determine which medication would work best for a patient’s condition using animal models. As a result, GEM models are less appropriate for systematic drug testing than xenograft models. First, GEM models are more costly and difficult to develop than xenograft models. Second, it might be challenging to breed mice to create GEM models with various genetic abnormalities. Third, because GEM models form tumors with a long latency and uneven penetrance, using them to evaluate treatment candidates on a broad scale presents difficulties. Fourth, patents and intellectual property rights frequently place limitations on utilizing publicly accessible GEM models. Fifth, pharmacological studies still need to test the majority of GEM models. The creation of clinically applicable mouse models that replicate the molecular, cellular, and genomic events of human cancers and clinical response is one of the future challenges in mouse modeling. Another is the creation of technologies that enable effective in vivo imaging and high-throughput screening in mice.

## 14. Scope and Significance

Multiple genetic mutations that change cells and enable their aberrant development, proliferation, and metastasis culminate in cancers. For advancements in diagnosis and treatment, these aberrations must be found and their role in the pathophysiology of cancer must be understood. Mice are an excellent choice for a model system because they are (a) tiny, (b) easy to keep, (c) breed quickly and have large litters, and (d) are amenable to genetic manipulation. To forecast new cancer indicators, distinguish between molecular prognostic biomarkers, and determine their function in the progression of the illness, the zebrafish cancer models are particularly suitable. Almost every form of the tumor with a shape and set of signaling pathways similar to those found in humans can arise spontaneously in zebrafish. The most important traits of zebrafish that make them ideal as a cancer model are their small size, large clutch size, low cost, ability to generate hundreds of embryos from a single mating, transparent embryos, and embryonic development outside the uterus.

## 15. Conclusions

Studies on mouse cancer models have shed important light on the molecular processes enabling tumor development, progression, metastasis, maintenance, and acquisition of chemoresistance. However, because mice and humans are two very different species, research on mouse models is unlikely to replace research on human cancer samples, cell lines, and patients. Studies on human cancer are helpful for identifying possible cancer genes and assessing anticancer drugs; however, they are heterogeneous and complicated, based on comparisons, and provide only minimal knowledge of the function of genes in carcinogenesis. They are not, however, the best models to test anticancer drugs. The particular processes that contribute to the formation of tumors may be understood using mice models, in contrast. The complexity of human malignancies must thus be understood using combinatorial methodologies as well as a systems approach that makes use of many model systems.

Zebrafish (Danio rerio) is a potential model to investigate human cancer. Researchers studying cancer are drawn to zebrafish because of their suitability for in vivo imaging, fast development, chemical screening, and adaptable genetics. The forward genetics and vertebrate biology of the zebrafish make it a model system with immense potential for understanding cancer. Mutant lines, xenotransplantation, transgenic lines, and chemical carcinogenesis are a few techniques that can be used to create tumors in zebrafish. Studies of tumor metastasis and invasion using zebrafish transplants can be very fruitful. Zebrafish may be used in cancer research in a variety of methods, including the analysis of -omics data using bioinformatics, the assessment of carcinogenesis, or the use of PDXs.

These models may be used to examine the processes of cancer development and progression, and they have been critical assets for the pre-clinical testing of a broad range of innovative medications and therapy approaches. Overall, based on the scientific problems that need to be addressed, this study offers a systematic and thorough discussion of how these various models may be applied.

The salient features of this review article with potential clinical relevance include:Mouse models can be effective platforms for confirming gene functions, discovering novel cancer genes, and evaluating potential anticancer drugs.GEM models have greatly improved our knowledge of the molecular processes that underlie tumor development, progression, metastasis, and chemotaxis. The use of mouse models has a number of drawbacks, including variances in tumor formation and medication response between different species.Genetically defined, homogenous tumors in mouse models can be a useful tool for finding biomarkers. With regard to tissue architecture, molecular characteristics, and treatment response, PDX models continue to bear striking resemblances to their tumors of origin.A system-level perspective employing an interdisciplinary ‘Team Medicine’ approach can provide a deeper understanding and novel treatment strategies.

## Figures and Tables

**Figure 1 cancers-15-02961-f001:**
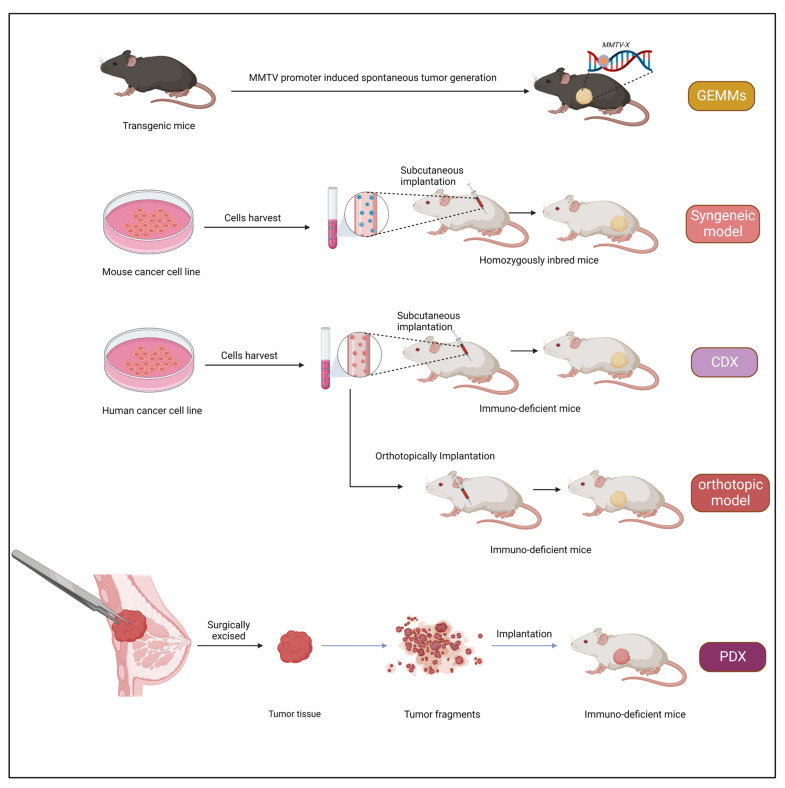
Schematic representation showing the generation of various mouse tumor models for cancer therapeutics.

**Figure 2 cancers-15-02961-f002:**
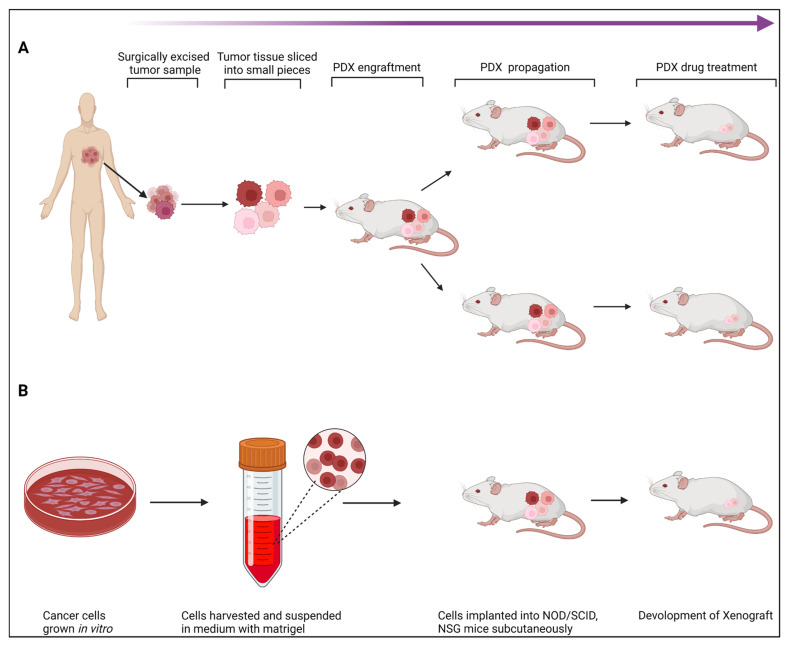
An overall schematic representation of patient-derived xenograft (PDX) generation. (**A**) Fragmentation of tumor tissue, followed by subcutaneous or orthotopic implantation, (**B**) Processing of tumor tissue as a single cell suspension, followed by subsequent implantation.

**Figure 3 cancers-15-02961-f003:**
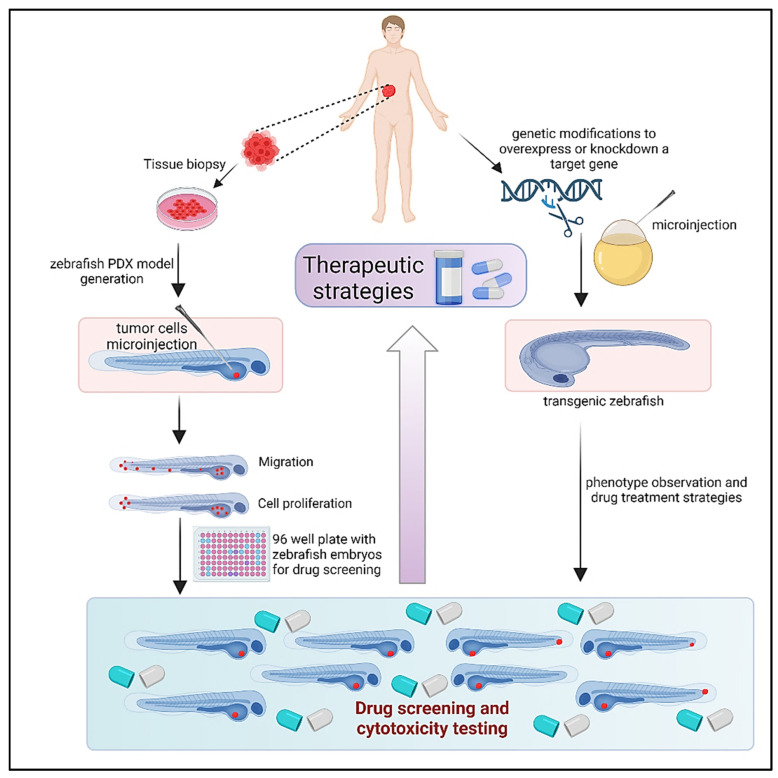
Generation and application of zebrafish models in cancer research. The tumor tissue obtained from patients is sectioned and processed and is transplanted into zebrafish larvae 48 h post-fertilization to create zPDXs. zPDX provides an effective platform for screening potential cancer drug candidates and evaluating cancer cell growth. Another major application of zebrafish is the generation of transgenic. The mutated DNA is microinjected into the zebrafish embryos for knockdown or overexpression of the disease target gene, followed by phenotypic observation.

**Table 1 cancers-15-02961-t001:** Benefits and limitations of various in vivo models used in cancer therapeutics.

Model	Benefits	Limitations	References
Syngeneic Model	Inexpensive Immunocompetent host Genetic manipulation is possible for the cells that undergo engraftment	Limited murine cells Does not apprehend tumor heterogenicity. Species-specific background	[16,17,18,19,20]
Xenograft Model	Genetic manipulation is possible for the cells that undergo engraftment. Inexpensive Reproducible	Lack of host immune response Limited cells tumor Does not apprehend tumor heterogenicity. Species-specific background	[16,17,18,21,22]
*nu/nu* (nude mouse, an immune deficient homozygous which lacks thymus and could not produce T-cells)	First immunodeficient mouse strain The total number of circulating lymphocytes is five to six times less in nude mice than in normal animals. Most of these cells are B cells, so they are used for numerous cancer metabolomics research. Highly correct prediction rates in comparison to in vitro systems for resistance and sensitivity of a tumor	A significant limiting factor is the duration of testing, as at least 4 months is needed before getting the test results. Nude mice are expensive, primarily due to the specialized breeding and maintenance required to maintain the phenotype. Limited utility for studying immune-based cancer therapies. Nude mice, lacking functional immune cells, cannot be used to evaluate the potential side effects and related toxicities. The absence of immune cells in nude mice alters the tumor microenvironment.	[23,24]
Severe combined immunodeficiency syndrome (SCID)	No mature B and T cells and decreased NK activity Provide realistic heterogeneity of tumor cells. It can predict the response of the drug against a tumor in human patients. It can allow the rapid analysis of human tumor response to a therapeutic regime.	Since they are immunocompromised, they provide a less realistic tumor microenvironment. They are expensive and technically complicated. Low level of engraftment of human cells They have a very short life span of approximately 8 months	[25,26]
Nonobese diabetic (NOD)- SCID gamma (NSG)	NSG mice live longer than any other immune-deficient mice Deficient in IL-2 receptor gamma chain and lack of mature B, T, NK cells, and cytosolic signaling. Used for metabolomics study for human immune deficiency virus	No primary immune response No multilineage hematopoiesis Expensive and technically complicated	[26]
Mouse PDX model	Preserve molecular and histological features. Mimic TME Appropriate for drug screening	Limited access to biological material Slow tumor growth Not so easy, technical expertise required	[27,28]
Human PDX model	Testing of new immunotherapies Mimic	Graft vs. host disease Expensive	[29,30,31,32]
GEMM	Testing of new immunotherapies Mimic	Graft vs. host disease	[33]
Zebrafish	Follow the process of tumor development. Low cost Drug screening	Not so easy. Skilled technical expertise is required. Drug screening Differ in physiological condition	[4,5,34,35]

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
