# Peer review of "Recent Advancement in Breast Cancer Research: Insights from Model Organisms—Mouse Models to Zebrafish"

_cancers, 2023, doi:10.3390/cancers15112961_

Round 1

Reviewer 1 Report

The article “ Recent Advancement in Breast Cancer Research: Insights from Model Organisms Mouse models to Zebrafish” is a review study about the functions of different animal models in cancer research developments. It discusses about different types of mic and zebrafish model with exploration of them. There is no evidence of method type and finally it underlines the important role of animal models in cancer treatment and progress in oncology research in future.

1. This is similar to be narrative review article more than being systematic review. There is no analysis of statistics data.

2. This is not a new subject to be studied in review article. However not enough quality.

Similar to : https://www.ncbi.nlm.nih.gov/pmc/articles/PMC7979343/

3. Keywords, page 1, please correct the font of written word mouse.

4. Please write the sources of figures.

5. Table 1, page 3, please write the references of data used in this table.

6. Table 1, page 4, figure 1, page 5, not right place. they must be after the related text.

7. No description of methods.

Author Response

Reviewer # 1

The article “Recent Advancement in Breast Cancer Research: Insights from Model Organisms Mouse models to Zebrafish” is a review study about the functions of different animal models in cancer research developments. It discusses about different types of mice and zebrafish models with exploration of them. There is no evidence of method type and finally it underlines the important role of animal models in cancer treatment and progress in oncology research in future.

  1. This is similar to be narrative review article more than being systematic review. There is no analysis of statistics data.

As indicated by an email conversation with the Section Managing Editor (Aiyarucht Techjindamanee) on May 3rd, 2023 (the day after submission), we were told this should be considered a regular review and NOT a systematic review.

  1. This is not a new subject to be studied in review article. However not enough quality.

Similar to: https://www.ncbi.nlm.nih.gov/pmc/articles/PMC7979343/

Compared to the reviewer’s mentioned article above on animal models, our article discussed not only the use of mouse and zebrafish models in cancer research but also discussed cancer therapeutics and a team medicine approach.

  1. Keywords, page 1, please correct the font of written word mouse.

            Corrected.

  1. Please write the sources of figures.

            YES, we mentioned it in the acknowledgments, as we did the figure ourselves using the Biorender tool.

  1. Table 1, page 3, please write the references of data used in this table.

            References have been included at the right margin of the Table.

  1. Table 1, page 4, figure 1, page 5, not right place. They must be after the related text.

Figures and Table are provided separately at the time of submission, and their positions/representations in the text will be decided by the Section Editor. We have not inserted the table or figures within the text of manuscript.

  1. No description of methods.

            Dear reviewer, we have already cited the appropriate references for the details.

Reviewer 2 Report

The systematic review is focusing on various animal models of breast cancer available today. The review is nicely written, however, there are some points to consider in order to improve the presentation. In particular, fixing typos and English errors, including in the main text and Tables, would improve the manuscript. 

1. Some abstracts are too long, e.g. page 2 out of 26 (first page of Introduction). Or page 8 out of 26; page 15. Etc. Shorter paragraphs make it easier to read the text.

2. Table 1. "Cells that undergoes" should be "cells that undergo". 

3. Table 1. This sentence is not complete "Nude mice are expensive they need special conditions behind laminar flow barriers to avoid infections"

4. Table 1 is mainly aligned, except for the following text (section) in the left column, starting with the "nu/nu (nude mouse)"

5. Table 1. SCID mice. "They have a very short life span of approximately 8 months" Statements like this need a reference. Although "very short" sounds too dramatic, it can be simply "short", given that the mice typically live up to 1.5-2.0 years, and most of the experiments are supposed to be done during the first year of life (except e.g. aging models). When fertile, 8 months is a good enough time to keep the line viable. 

6. Overall, Table 1 is informative and useful, although it requires alignment and proofreading.

7. Figure 1 is nice-looking and informative. It would benefit from aligning the text and making the arrow size more concise, e.g. in the lower section.

8. Consider using italics consistently, e.g. for "in vivo", "in vitro".

9. Consider to use consistent genetic terms, i.e. "knock-out and knock-in" or "knockout and knockin"

10. Page 8. "at -80 oc" should be "at -80 oC" with capital "C".

11. Page 13. Fix temperature signs:  36 oC,  28 oC.

12. Table 1 would benefit from references included in the Table text to support the statements.

13. Only 20-25% of references are recent. The review would benefit from prioritizing literature from the recent 5 years.

There are several typos, errors, and inconsistencies that need to be fixed during the proofreading and revision. I provided some examples in the comments, but there are more places to fix in the text.

Author Response

Reviewer # 2

The systematic review is focusing on various animal models of breast cancer available today. The review is nicely written, however, there are some points to consider in order to improve the presentation. In particular, fixing typos and English errors, including in the main text and Tables, would improve the manuscript. 

We sincerely appreciate the reviewer’s positive consideration and constructive comments regarding the manuscript. We have extensively reviewed and revised the manuscript to ensure grammatical corrections and clarity of comprehension.

  1. Some abstracts are too long, e.g. page 2 out of 26 (first page of Introduction). Or page 8 out of 26; page 15. Etc. Shorter paragraphs make it easier to read the text.

I agree, however, due to the complexity of the subject, we felt that these paragraphs are very important and could not be shortened further before significant meaning is lost. We appreciate your consideration. 

  1. Table 1. "Cells that undergoes" should be "cells that undergo". 

            Corrected.

  1. Table 1. This sentence is not complete "Nude mice are expensive they need special conditions behind laminar flow barriers to avoid infections."

Corrected.

  1. Table 1 is mainly aligned, except for the following text (section) in the left column, starting with the "nu/nu (nude mouse)"

            We agree with the reviewer’s concerns which have been corrected in the revised manuscript.

  1. Table 1. SCID mice. "They have a very short life span of approximately 8 months" Statements like this need a reference. Although "very short" sounds too dramatic, it can be simply "short", given that the mice typically live up to 1.5-2.0 years, and most of the experiments are supposed to be done during the first year of life (except e.g. aging models). When fertile, 8 months is a good enough time to keep the line viable. 

            We agree with the reviewer’s concerns which have been corrected in the revised manuscript.

  1. Overall, Table 1 is informative and useful, although it requires alignment and proofreading.

            We sincerely thank the reviewer for the kind comments.

  1. Figure 1 is nice-looking and informative. It would benefit from aligning the text and making the arrow size more concise, e.g. in the lower section.

            We sincerely thank the reviewer for the kind comments.

  1. Consider using italics consistently, e.g. for "in vivo", "in vitro".

We apologize for some obvious errors.  Now, the revised manuscript has been edited accordingly.

  1. Consider to use consistent genetic terms, i.e. "knock-out and knock-in" or "knockout and knockin"

We apologize for some obvious errors.  Now, the revised manuscript has been edited accordingly.

  1. Page 8. "at -80 oc" should be "at -80 oC" with capital "C".

Corrected.

  1. Page 13. Fix temperature signs:  36 oC, 28 oC

Corrected.

  1. Table 1 would benefit from references included in the Table text to support the statements.

            References have been included at the right margin of the Table.

  1. Only 20-25% of references are recent. The review would benefit from prioritizing literature from the recent 5 years.

            New references from recent years have been included in the revised manuscript.

Reviewer 3 Report

This is a very intriguing review. The topic is interesting; it is well-organized and the figures are well-performed.

Questions and/or corrections:

There are some typos in the text and some verbs that are not correctly accorded. 

A table resuming limitations should be added.

The major questions is:

In addition to all the models presented, breast cancer is able to spread and generate metastasis also by perineural invasion. Perineural invasion (PNI) is a pathologic finding observed across a spectrum of solid tumours, typically with adverse prognostic implications. 

Please comment and discuss these papers  https://doi.org/10.1186/s13578-023-01008-4; https://doi.org/10.1074/jbc.M010499200; https://doi.org/10.3389/fcell.2021.676568.; https://doi.org/10.1038/s41598-020-60030-5.

and describe if there are in vivo models to study the PNI generated by breast cancer cells, if there is a necessity and what are the limits and benefits

There are only a few typos and mistakes. It is good. 

Author Response

Reviewer # 3

This is a very intriguing review. The topic is interesting; it is well-organized, and the figures are well-performed.

We appreciate the reviewer’s assessment and sincerely thank you for the kind comments.

Questions and/or corrections:

There are some typos in the text and some verbs that are not correctly accorded. 

            We apologize for some obvious errors.  Now, the revised manuscript has been edited accordingly.

A table resuming limitations should be added.

Table 1 has been given in the manuscript highlighting the benefits and limitations of various in vivo models used in cancer therapeutics.

The major questions is:

In addition to all the models presented, breast cancer is able to spread and generate metastasis also by perineural invasion. Perineural invasion (PNI) is a pathologic finding observed across a spectrum of solid tumors, typically with adverse prognostic implications. 

Please comment and discuss these papers  https://doi.org/10.1186/s13578-023-01008-4; https://doi.org/10.1074/jbc.M010499200; https://doi.org/10.3389/fcell.2021.676568.; https://doi.org/10.1038/s41598-020-60030-5  and describe if there are in vivo models to study the PNI generated by breast cancer cells if there is a necessity and what are the limits and benefits.

The detailed summary of perineural invasion, involving different in vivo models has been included with related benefits and limitations. A dialogue on case studies of research papers (link highlighted above) on perineural invasion has also been inculcated and suitable references have been cited in the text as per reviewer’s comment.

Round 2

Reviewer 1 Report

I dont have more comment.